# The Molten Globule, and Two-State vs. Non-Two-State Folding of Globular Proteins

**DOI:** 10.3390/biom10030407

**Published:** 2020-03-06

**Authors:** Kunihiro Kuwajima

**Affiliations:** 1Department of Physics, School of Science, the University of Tokyo, 7-3-1 Hongo, Bunkyo-ku, Tokyo 113-0033, Japan; kuwajima@ims.ac.jp; Tel.: +81-90-5435-6540; 2School of Computational Sciences, Korea Institute for Advanced Study (KIAS), Seoul 02455, Korea

**Keywords:** protein folding, molten globule state, two-state proteins, non-two-state proteins

## Abstract

From experimental studies of protein folding, it is now clear that there are two types of folding behavior, i.e., two-state folding and non-two-state folding, and understanding the relationships between these apparently different folding behaviors is essential for fully elucidating the molecular mechanisms of protein folding. This article describes how the presence of the two types of folding behavior has been confirmed experimentally, and discusses the relationships between the two-state and the non-two-state folding reactions, on the basis of available data on the correlations of the folding rate constant with various structure-based properties, which are determined primarily by the backbone topology of proteins. Finally, a two-stage hierarchical model is proposed as a general mechanism of protein folding. In this model, protein folding occurs in a hierarchical manner, reflecting the hierarchy of the native three-dimensional structure, as embodied in the case of non-two-state folding with an accumulation of the molten globule state as a folding intermediate. The two-state folding is thus merely a simplified version of the hierarchical folding caused either by an alteration in the rate-limiting step of folding or by destabilization of the intermediate.

## 1. Introduction

Elucidation of the molecular mechanisms of protein folding is a fundamental problem in biophysics and molecular biology, although almost 60 years have elapsed since Anfinsen and his coworkers discovered the genetic control of the native tertiary structure of proteins [1]. In the interim, there have been many experimental studies on protein folding (refs. [2,3,4] and references cited therein). From these studies, it is now clear that there are two types of folding behavior, i.e., non-two-state folding, involving at least one folding intermediate, and two-state folding without any detectable intermediate during kinetic refolding from the fully unfolded (U) state to the native (N) state [5,6]. Th thermodynamic states, like N, U, and the folding intermediate, of a protein surrounded by solvent water are large ensembles of many microstates, but the different thermodynamic states are well separated from each other by a free-energy barrier higher than the thermal fluctuation energy. Understanding the relationships between the two-state and non-two-state folding may thus be essential in order to fully elucidate the molecular mechanisms of protein folding. This article will discuss these relationships on the basis of available data on the correlations of the rate constant of folding with various structure-based properties, which are determined primarily by the backbone structure (topology) of proteins.

If we assume that a protein folds into the N state by a random search of all possible conformations, the time required for successful folding would be unrealistically long for even a small protein. This is known as Levinthal’s paradox [7] (see also [2,8]). Traditionally, the presence of a specific pathway of folding was thought to provide a specific and unique way to get from U to N in a reasonable amount of time [7], and hence, detection and characterization of specific intermediates, present along the pathway of folding, were basic approaches for elucidating the mechanisms of folding. The molten globule (MG) state was proposed as such a specific intermediate of folding [9], and experimentally, the MG state was observed as an equilibrium unfolding intermediate in certain proteins, and also as a transient folding intermediate, which accumulates at an early stage of kinetic refolding [10,11]. The transient MG state was observed not only in the proteins that show the equilibrium MG state but also in the proteins that show the two-state unfolding transition at equilibrium. Oleg B. Ptitsyn, our colleagues, and I thus conducted a cooperative research project supported by the Human Frontier Science Program (HFSP) for three years from 1993. The title of the project was “Role of the Molten Globule State in the Folding of Globular Protein.” In the next section (Section 2) of this article, the MG state in protein folding is thus described.

The two-state folding behavior, in which both equilibria and kinetics of folding/unfolding occur only between N and U, was discovered for a number of small globular proteins, usually with less than 100 amino acid residues [12,13], and this discovery changed the focus of many researchers in regard to the two-state proteins. These initial efforts suggested that the two-state folding, which required no stable intermediates for complete and successful folding to the N state, was simpler than the non-two-state folding. Theoretical and computational studies of protein folding have proposed an energy landscape theory (a funnel model) of folding, and the two-state folding behavior has been interpreted reasonably well by this model [14]. In Section 3, the two-state vs. non-two-state folding of globular proteins will be described. This section mainly deals with the significance of a burst-phase intermediate, which is accumulated within the dead time of kinetic folding experiments, because the burst-phase intermediate has often been observed in non-two-state proteins [11]. Sub-millisecond continuous-flow techniques [15] and more recently in-gel folding techniques [16] have revealed that the burst-phase intermediate is a real folding intermediate separated from the U state by a free-energy barrier, so that there are two-types of globular proteins in terms of the folding behavior, i.e., two-state proteins and non-two-state proteins.

In Section 4, the relationships between the two-state and non-two-state folding reactions will be discussed. The logarithmic rate constant of folding, ln(*k*_f_), shows significant correlations with structure-based properties, which are determined by the backbone structure. Recently, various structure-based properties that correlate with ln(*k*_f_) were reported, and the correlations of ln(*k*_f_) with the structure-based properties were found to be quite similar between two-state and non-two-state proteins, indicating that the physical mechanisms behind two-state and non-two-state folding are essentially identical (see Section 4). From these results, it is concluded that the general mechanism of protein folding can be described using a two-stage hierarchical model, with the two-state folding being merely a simplified version of the hierarchical folding.

## 2. Role of The Molten Globule State in Protein Folding

The molten globule (MG) state is an intermediate ensemble between the N and the U states of globular proteins. It has the following characteristics: (1) the presence of a substantial amount of native-like secondary structure; (2) the absence of most of the specific tertiary structure associated with the tight packing of side chains; (3) the presence of a native-like backbone fold (topology), and the compactness of the overall shape of the molecule, with a radius only 10%–30% larger than that in the N state; and (4) the presence of a loosely organized hydrophobic core, and the heterogeneity of the three-dimensional structure, in which certain subdomains of the molecule are more organized than others [10,11]. For more recent and comprehensive information on the MG state, see [17]. The loosely organized hydrophobic core is often characterized by binding experiments using a hydrophobic fluorescent probe, 1-anilino-8-naphthalene sulfonate (ANS) [18]. The equilibrium MG state has been observed for a number of globular proteins under mildly denaturing conditions (e.g., at acidic pH, at a high pressure or in the presence of an intermediate concentration of a strong denaturant such as guanidinium chloride (GdmCl) or urea) [10,11,19]. The role of the MG state in kinetic folding of globular proteins was an intriguing issue when our project started [20], and Oleg proposed that the MG state is a general intermediate in protein folding [9]. In fact, transient kinetic intermediates were detected and characterized for a number of globular proteins, including α-lactalbumin [21], lysozyme [21,22], cytochrome *c* [23], ribonuclease A [24] and apo-myoglobin [25], by a kinetic circular dichroism (CD) technique and a pulsed hydrogen/deuterium (H/D)-exchange method combined with two dimensional (2D) NMR spectroscopy. Close similarity between the equilibrium MG state and the kinetic folding intermediate thus characterized was demonstrated for certain proteins by coincidence of the equilibrium unfolding transition curve of the MG state and the pre-equilibrium unfolding transition curve of the kinetic intermediate [26,27], and by close similarity of the H/D-exchange protection profile between the equilibrium and kinetic intermediates [25]. 

To further characterize the equilibrium and kinetic MG states, Ptitsyn’s group and the Japanese members of our cooperative project carried out joint experiments, and Gennady V. Semisotnov (Institute of Protein Research, Russia) often visited Japan to participate in these efforts. Hiroshi Kihara (Kansai Medical Univ.), Yoshiyuki Amemiya (Univ. Tokyo), Kazumoto Kimura (Dokkyo Univ.), and students of our laboratories at that time were also involved in these experiments. We utilized a synchrotron radiation facility at the High Energy Accelerator Research Organization, Tsukuba, Japan to characterize the MG state of proteins by a small angle X-ray scattering (SAXS) technique, which gives us the information about the size and shape of a protein molecule in solution [29]. Figure 1 shows small-angle X-ray scattering (SAXS) patterns and Kratky plots of native and fully unfolded carbonic anhydrase [28]. The SAXS patterns are represented by the scattering intensity, *I*(*h*), as a function of the scattering vector, *h*, where *h* = (4πsinθ)/λ; 2*θ* is the scattering angle, and *λ* is the wavelength of the X-ray. The Kratky plots are given by *I*(*h*) × *h*^2^ as a function of *h* [29]. From Figure 1, we can see large differences in the scattering intensity and the shape of scattering curves between the globular N state and the coil-like U state of the protein [28]. Such differences in the scattering properties encouraged us to investigate the time-resolved SAXS during kinetic refolding. If the early kinetic folding intermediate is identical to the MG state, it would be possible to directly observe a compact shape of the protein molecule, an important characteristic of the MG state, at an early stage of kinetic folding. We investigated the kinetic refolding reactions of β-lactoglobulin and α-lactalbumin induced by a denaturant (urea or GdmCl) concentration jump using a stopped-flow SAXS apparatus [30,31]. Both the proteins formed a compact globular structure with a radius of gyration (*R*_g_) identical to that of the MG state, although the intermediate of β-lactoglobulin exhibited a non-native α-helical structure [30,32].

Figure 2 shows time-dependent changes in the *R*_g_ value during refolding of α-lactalbumin and the Kratky plots of the protein in the early stages of refolding [31]. Changes in the *R*_g_^2^ values were fitted to a single-exponential function, because *R*_g_^2^ (instead of *R*_g_) has a linear dependence upon the fractional populations of the individual states [31,33]. The rate constant thus obtained was 0.49 (±0.07) s^−1^, which is in agreement with the rate measured by other spectroscopic techniques. The *R*_g_ value after complete refolding was 15.5 (±0.1) Å, and the *R*_g_ value at zero time of refolding obtained by extrapolation of the fitting curve to zero time was 17.5 (±0.2) Å. Because this protein forms the transient folding intermediate within the dead time (~7 ms) of the stopped-flow measurement, the folding intermediate has an *R*_g_ of 17.5 (±0.2) Å, which is in good agreement with the known *R*_g_ value of the equilibrium MG state [31]. The Kratky plot obtained by averaging the scattering pattern within 10–20 ms of refolding is coincident with that of the equilibrium MG state, indicating that both the size and shape of the transient folding intermediate were identical to those of the equilibrium MG state.

In the SAXS experiments of Figure 2, we used a 2D charge-coupled device (CCD)-based X-ray detector with a beryllium-windowed X-ray image intensifier, by which the S/N ratio of the scattering data was dramatically improved as compared with the data obtained by a one-dimensional detector [31]. This 2D CCD-based detector was developed by Amemiya and coworkers [34], and detailed procedures for data correction and the analysis software were described by Ito et al. [35]. The 2D CCD-based detector was used for nearly a decade in the early 2000s to characterize compact folding intermediates of other globular proteins [36,37,38,39,40,41,42], but at present, a more sensitive PILATUS pixel detector is widely used for the SAXS experiments [43,44,45].

## 3. Two-State vs. Non-Two-State Folding of Globular Proteins

In 1991, just two years before the start of the above-mentioned project supported by HFSP, Jackson and Fersht [12] in Cambridge reported that barley chymotrypsin inhibitor 2 (CI2), a small monomeric protein of 83 residues (the first 18 residues are unstructured), appears to be a rare example in which both equilibria and kinetics are described by a two-state model. As far as I know, this was the first report on observation of the two-state folding of globular proteins. The criteria for the two-state folding are (i) observation of a single equilibrium unfolding transition as a function of an external parameter such as denaturant (GdmCl or urea) concentration or temperature; (ii) linear free-energy relationships of the equilibrium unfolding free energy and the activation free energies for kinetic folding and unfolding with respect to the denaturant concentration; (iii) single-exponential kinetics for both folding and unfolding with the observation of the full change of a structural probe, fluorescence, CD ellipticity or any other index used to monitor the conformational transition, between U and N; (iv) consistency between the equilibrium and the kinetic parameters of folding/unfolding; and (v) agreement between the van’t Hoff enthalpy and the calorimetric enthalpy of equilibrium thermal unfolding. Any deviations from the above criteria suggest the presence of an intermediate between N and U [2]. However, when a proline peptide bond undergoes a slow *cis-trans* isomerization in the U state, the folding kinetics become complex in spite of the absence of any intermediates between N and U [2,46], and such a case may also be classified as a two-state folding.

Although Jackson and Fersht [12] reported that CI2 was a rare example, later many small proteins, usually with less than 100 amino-acid residues, were shown to fold with a similar simple two-state mechanism, and in 1998, Jackson reported more than 20 examples of two-state folders, which show wide variation in folding rates from microseconds to seconds [13]. Interestingly, the kinetic refolding reaction of protein L, a typical two-state protein, measured by the time-resolved SAXS has shown that the chain collapse occurs concomitantly with the formation of the N state, although there is some disagreement between SAXS and single-molecule Förster resonance energy transfer experiments [33,47]. My colleagues in KIAS (the Korea Institute for Advanced Study) and I recently constructed a standardized protein folding database (PFDB) with temperature correction (http://lee.kias.re.kr/~bala/PFDB/), in which the ln(*k*_f_) values of all listed proteins were calculated at the standard temperature (25 °C) [48]. Among the total 141 proteins registered in PFDB, 89 are two-state proteins, and hence two-state proteins are no longer rare examples. However, this does not necessarily mean that a majority of naturally occurring proteins are of the two-state type. Because two-state proteins are apparently simpler than non-two-state proteins, researchers in the protein folding field tend to investigate two-state proteins, which would account for the large number of two-state proteins registered in PFDB. Among the 89 two-state proteins registered in PFDB, 53 are composed of an isolated domain of a multi-domain protein. The average chain length of the two-state proteins registered in PFDB is 92.6, while the average chain length of the non-two-state proteins is 139.4 [48].

The folding intermediate appears not to be required for the complete folding of two-state proteins, which has raised questions as to the significance of the folding intermediates previously detected and characterized in non-two-state proteins [49,50]. The formation of an early kinetic intermediate often occurs too quickly to measure directly by a conventional rapid reaction technique such as the stopped-flow method, and the process occurs in a burst phase within the dead time of the measurement [11,51,52]. This situation raises a question in regard to the burst-phase—namely, whether an observed burst-phase signal reflects a real folding event. To answer this question, the burst-phase fluorescence or CD signals at an early stage of kinetic refolding of cytochrome *c* and ribonuclease A were thus investigated by Sosnick et al. [53,54,55,56]. The refolding reactions were induced by stopped-flow concentration jumps of GdmCl, and the burst-phase signals thus obtained were compared with the signals of non-folding polypeptides: fragments 1–80 and 1–65 for cytochrome *c* and a disulfide broken form for ribonuclease A. Rather surprisingly, the signals of the intact proteins and the non-folding polypeptides were coincident with each other. It was thus concluded that the sub-millisecond burst-phase signals may not reflect the fast formation of the folding intermediate, but rather may reflect some solvent-dependent contraction/collapse of the U state, since the water solution at a low concentration of GdmCl is a poor solvent for the polypeptides [53,54,55], and the initial barrier mechanism in protein folding was proposed [56]. Hydrophobic amino-acid residues, usually buried inside the molecule in the N state, are exposed to solvent water in the U state, and hence the water solution is thought to be a poor solvent for the unfolded polypeptide [57].

Roder’s group [58,59] and Takahashi’s group [39,60] thus investigated the sub-millisecond processes of folding reactions of cytochrome *c* and ribonuclease A by continuous-flow techniques, which have a higher time resolution than the conventional stopped-flow method. In 1998, Shastry and Roder [58] found that the kinetics of the cytochrome *c* folding measured by tryptophan (Trp 59) fluorescence exhibits a major exponential process with a time constant of ~60 μs, independent of initial conditions and heme ligation state, indicating that a common free energy barrier is encountered during the formation of an early compact intermediate. Therefore, there are at least two distinct stages in the cytochrome *c* folding, with the compact folding intermediate accumulating in the first stage, followed by the rate-limiting formation of the N state. Interestingly, Qui et al. [61] found by the laser temperature-jump method that not only full-length cytochrome *c* but also non-folding peptides 1–65 and 1–80 displayed similar exponential decays, indicating that the free-energy barrier to collapse was similar in the folding and non-folding sequences. Because the non-folding peptides contain significant portions of the natural cytochrome *c* sequence, it might be possible that such sequences can form a similar compact structure [62]. For ribonuclease A, Welker et al. [59] reported that the very fast folding U state (U_vf_), which contains only native proline isomers, exhibits an exponential decay in fluorescence with a time constant of ~80 μs, whereas the equilibrium U state (U_eq_), which contains nonnative proline isomers, did not show this exponential decay, indicating that the isomerization state of proline peptide bonds can have a major impact on the early stages of the kinetic folding. It may be noted that the burst phase previously observed in ribonuclease A by Houry et al. [52] using a stopped-flow double-jump apparatus was for U_vf_ and not for U_eq_. Kimura et al. [39] have shown that the sub-millisecond folding intermediate of intact ribonuclease A strongly binds to ANS and has a compact structure, characteristics of the MG state, but that disulfide-reduced ribonuclease A shows neither any significant binding to ANS nor such compaction of the structure, indicating the difference in the sub-millisecond conformations of ribonuclease A and its reduced form.

Recently, Okabe et al. [16] employed an elegant technique to delineate the solution burst-phase events of folding by encapsulating the proteins in silica gels. Within a porous silica gel with a large water content, large scale protein motions are dramatically slowed, but the in-gel N state can retain its solution properties [63,64,65,66]. In particular, the folding of a protein in such a gel is dramatically slowed, and this makes it possible to directly observe the entire folding process without substantially perturbing the folding pathway [67,68,69,70]. Figure 3 shows (A) representative far-UV CD spectra of cytochrome *c* during the refolding and (B) a typical kinetic refolding curve measured by the CD ellipticity at 220 nm for the initial resolvable phase of the in-gel refolding; the protein that was initially acid unfolded at pH 1.8 was refolded at pH 4.5 and 25°C in a silica gel containing 80% (*w*/*w*) water [16]. The CD spectroscopy in the far-UV region provides a very effective tool to characterize the secondary structure of a protein, and in particular, the CD ellipticity at 220 nm gives a measure of the α-helical content of the protein [71]. The CD spectra in the N and U states of cytochrome *c* in the gel are nearly identical to the corresponding spectra in solution, indicating that the conformations in the N and U states are not influenced by the gel matrix (Figure 3A). As shown in Figure 3B, the initial phase of the in-gel folding exhibits an exponential decay with a time constant of ~30 s. Because the fastest phase of the protein in solution at pH 4.5 and 22 °C has a time constant of 57 μs [58], the refolding rate was at least 5.3 × 10^5^ times reduced in the in-gel folding compared with the refolding in solution. The CD spectrum of the early kinetic intermediate observed at 3 min, 6 times the time constant (~30 s), in the in-gel folding (Figure 3A) is very similar to the CD spectrum at 400 μs, seven times the time constant (57 μs), in the free solution refolding reported by Akiyama et al. [60], suggesting that the same transient folding intermediate accumulates at the initial stage in the in-gel and the free solution refolding reactions. It may also be noted that the acid compact state of cytochrome *c* stabilized by KCl, known also as the MG state [72,73], is an equilibrium counterpart of a late folding intermediate separated from U by the rate-limiting free-energy barrier [74], and hence it is different from the intermediate shown in Figure 3 and the MG state in this article.

Figure 4 shows kinetic refolding curves measured by the CD ellipticity at 220 nm for the initial resolvable phase of the in-gel refolding reactions of four proteins, equine β-lactoglobulin, human tear lipocalin, bovine α-lactalbumin, and hen-egg white lysozyme [16]. The proteins were initially unfolded in the silica gel immersed into 6 M GdmCl, and the refolding was initiated in the gel by a GdmCl concentration jump produced by washing the protein-embedded gel with a refolding buffer solution. From Figure 4, all four proteins exhibit an exponential decay and accumulate a transient folding intermediate. The rate constants measured at different wavelengths were coincident with each other (insets of Figure 4), and the CD spectrum of the transient intermediate for each protein was obtained from the kinetic refolding curves measured at different wavelengths. From comparisons of the transient CD spectrum thus obtained with the CD spectrum of the burst-phase intermediate in free refolding in solution and also with the CD spectrum of the equilibrium MG state, if available, for each protein, it is concluded that the folding intermediate formed at the fastest initial stage in the in-gel refolding is equivalent to the intermediate formed in the burst phase in the free refolding in solution. Figure 4 also shows that the CD ellipticity value for each protein obtained by extrapolation of the exponential decay curve to zero time does not coincide with the value in the U state, indicating the presence of a burst phase before the exponential decay. However, the U state values shown in Figure 4 are those at 6 M GdmCl, and when we assume a linear dependence of the U state values on GdmCl concentration, the zero-time ellipticity becomes much closer to the values in the U state under the refolding condition. The burst-phase signals in Figure 4 may represent the rapid collapse of the protein molecules caused by changing the solvent environment from a good solvent (6 M GdmCl) to a poor solvent (~0 M GdmCl). 

## 4. The Relationships between the Two-State and Non-Two-State Folding Reactions

From the above results, it now appears that there are two types of globular proteins in terms of the folding behavior, i.e., two-state proteins and non-two-state proteins. The two-state folding reactions under native conditions are simply represented by Equation (1), where the backward reaction is neglected because it is negligible as compared with the forward reaction:(1)$$U\overset{k_{f}}{\to }N$$

On the other hand, the non-two-state proteins often accumulate the MG state at an early stage in kinetic refolding from the U state [11]. The simplest reaction scheme of the non-two-state folding is thus given by:(2)$$U\overset{k_{\mathrm{UI}}}{\underset{k_{\mathrm{IU}}}{⇌}}I\overset{k_{f}}{\to }N$$
where I is the folding intermediate that has the characteristics of the MG state, *k*_UI_ is the microscopic rate constant of formation of I from U, and *k*_IU_ is the microscopic rate constant of formation of U from I. For the non-two-state proteins that accumulate a sufficiently stable I state during folding, *k*_UI_ is much larger than *k*_IU_, and hence, *k*_UI_ is eventually equivalent to the apparently observed rate constant of formation of I; this observed rate constant is denoted by *k*_I_ in the following. Understanding the relationship between the two types of proteins may be important for fully elucidating the molecular mechanisms of protein folding. Although Equations (1) and (2) represent the simplest schemes for the two-state and the non-two-state folding with a single defined pathway, the folding reactions along multiple parallel pathways have also been observed in real proteins [2,75], but the arguments and conclusions in this article may be valid also for the multiple-pathway folding.

For two-state proteins, significant correlations between ln(*k*_f_) and various structure-based properties have been well documented in the late 1990s and the early 2000s. The structure-based properties include the relative contact order (RCO) [76,77,78], the long-range order (LRO) [79,80], the number of sequence-distant native pairs (*Q*_d_) [81,82], the chain topology parameter [83], the total contact distance [84], the cliquishness [85] and the relative logarithmic contact order [86]. All these structure-based properties are determined by the number of contacts between two residues in close proximity, with their C_α_–C_α_ distance less than 6–8 Å, and by the separation lengths between the contacting residues along the primary sequence. These structure-based properties thus depend on the backbone structure (topology) of a protein, with a more complex backbone topology corresponding to a slower the folding rate. Originally, it was proposed that the correlations of ln(*k*_f_) with the above structure-based properties (except cliquishness) are present only for two-state proteins, and for non-two-state proteins, whose folding reactions may be complicated by the rate of escape from the folding intermediates trapped on a ragged energy landscape of folding, such simple correlations of ln(*k*_f_) with the structure-based properties may be absent [82].

However, later it became clear that non-two-state proteins also show similar correlations with the structure-based properties. The ln(*k*_f_) values of non-two-state proteins show a significant correlation with the chain length (*L*), i.e., the number of amino acid residues, and also with the absolute contact order (ACO), which is given by ACO = RCO × *L* [87,88,89]. Similar correlations of ln(*k*_f_) with ACO were observed for two-state and non-two-state proteins, but the correlation with *L* was worse for two-state proteins than for non-two-state proteins [87,89]; see also [90]. Among the above-mentioned structure-based properties for two-state proteins, three properties, LRO, *Q*_d_ and the cliquishness, were also reported to exhibit significant correlations for non-two-state proteins [85,89,91]; LRO and *Q*_d_ are closely related as LRO ≈ *Q*_d_/*L*. More sophisticated structure-based properties, well correlating with ln(*k*_f_) for both two-state and non-two-state proteins, have also been proposed [92,93,94,95,96,97,98,99,100]. These structure-based properties include the *n*-order contact distance [93], the geometric contact number, which is the number of nonlocal contacts well packed by a Voronoi criterion [94], the inter-residue interaction parameter, which considers the distances between all residue pairs in a protein [97], the average topological information, which is derived from probability densities of all residue pairs based on a self-avoiding random walk model [99], the entanglement of the native backbone structure [98], the logarithmic ACO, computed on shortcut networks of the native structure [100], the prediction by an algorithm for the prediction of folding and unfolding rates (PREFUR) using *L* and the structural class as only protein-specific input [96], and the effective chain length, evaluated from the number of helical residues and the number of helices [92] or from the amino acid composition of a protein [95]. More strict physical models to evaluate ln(*k*_f_) for two-state and non-two-state proteins from the free-energy profiles obtained by simple statistical thermodynamic calculations were also reported [101,102,103,104]. Garbuzynskiy et al. [105] have shown that physical theory, which describes scaling relationships between *L* and the activation free energy of folding, together with biological constraints outlines a “golden triangle” limiting the possible range of ln(*k*_f_) for single-domain globular proteins. The ln(*k*_f_) values of most of the globular proteins, including both two-state and non-two-state folders, are distributed within this narrow triangle.

Figure 5A shows the dependence of the logarithmic rate constant of folding on *Q*_d_ for two-state and non-two-state proteins, taken from Kamagata et al. [89]. *Q*_d_ in Figure 5A represents the number of sequence-distant native pairs, where the two residues in each pair are separated more than 12 residues along the primary sequence, and the C_α_– C_α_ distance in space of the two residues is within 6 Å in the PDB structure of the protein [82]. Following the topomer search model of folding, proposed by Makarov and Plaxco [82], the logarithm of (*k*_f_/*Q*_d_) is plotted against *Q*_d_ in Figure 5A, and the ln(*k*_f_/*Q*_d_) values are shown for 18 two-state proteins and 22 non-two-state proteins. Among the 22 non-two-state proteins, the *k*_I_ values were also determined experimentally for 10 proteins, and hence, the values of ln(*k*_I_/*Q*_d_) are also presented for these proteins. Among the remaining 12 non-two-state proteins, seven exhibited the burst-phase intermediate, and five showed only a roll-over in the chevron plot; see [2] for comprehensible descriptions of the roll-over and the chevron plot. From Figure 5A, both the value of ln(*k*_f_/*Q*_d_) and its dependence on *Q*_d_ in the case of the two-state proteins are very similar to those found in the plots of ln(*k*_I_/*Q*_d_) and ln(*k*_f_/*Q*_d_) against *Q*_d_ for the non-two-state proteins. Such similarities between two-state and non-two-state proteins were also found rather generally in the relationships between ln(*k*_f_) and structure-based properties (see Figure 5B), which include ACO [87,88,89], LRO [91], the cliquishness [85], the *n*-order contact distance [93], the geometric contact number [94], the inter-residue interaction parameter [97], the average topological information [99], the entanglement of the native backbone structure [98], and the other structure-based properties [92,95,96,100]. As to the golden triangle for scaling of protein folding rates, proposed by Garbuzynskiy et al. [105], we also find similar distributions of ln(*k*_f_) between the two-state and the non-two-state proteins. These similarities thus clearly demonstrate that the physical mechanisms of folding are not essentially different between the two types of proteins. 

If the mechanisms behind two-state and non-two-state folding are essentially identical, what accounts for the differences between the two-state and the non-two-state proteins? Figure 5A, which includes the data for both *k*_f_ and *k*_I_ for non-two-state proteins, may provide a reasonable answer to this question. From Figure 5A, the slope of the plot of ln(*k*_f_/*Q*_d_) against *Q*_d_ is steeper (i.e., more negative) for the two-state proteins than for the non-two-state proteins. This trend can also be found in the plots of ln(*k*_f_) against the other structure-based properties, which usually represent the backbone topological complexity of a protein (Figure 5B) [93,94,97]. The steeper dependence of the plots against *Q*_d_ and the other structure-based properties for the two-state proteins may be interpreted in terms of a shift in the rate-limiting step, caused by a change in the value of each structure-based property. From Figure 5A, the *k*_f_ values of the two-state proteins are larger than those of the non-two-state proteins at *Q*_d_ values less than 25, but rather coincide with the *k*_I_ values of the non-two-state proteins. This apparently suggests that the U→I process of Equation (2) has become rate-limiting in these two-state proteins. At *Q*_d_ values between 25 and 70, the two-state *k*_f_ shows variations in either the non-two-state *k*_I_ or *k*_f_, and at *Q*_d_ values larger than 70, the two-state *k*_f_ values rather coincide with the non-two-state *k*_f_ values. It is thus suggested that the I→N process of Equation (2) is now rate-limiting in the two-state proteins with a *Q*_d_ larger than 70. 

From the above results, a two-stage hierarchical model is proposed as a general mechanism of protein folding. In this model, protein folding occurs in a hierarchical manner, reflecting the hierarchy of the native three-dimensional structure, as embodied in the case of non-two-state folding with an accumulation of the MG state as a folding intermediate. In this model, the protein folding process is divided into two stages: the first stage is formation of the MG state (U→I in Equation (2)); and the second stage is formation of the N state from the MG state (I→N). The two-state folding is thus merely a simplified version of the hierarchical folding caused either by an alteration in the rate-limiting step of folding or by destabilization of the intermediate. Therefore, there are two scenarios for interpreting the two-state folding behavior. First, when the backbone topology is sufficiently simple and organized by contacts between residues nearby in the primary sequence, the protein can easily form the specific conformation of side-chain packing, because the number of possible specific interactions is so small that the formation of the backbone structure rather uniquely determine the side-chain packing interactions. In this case, the first stage of the hierarchical folding becomes rate-limiting, leading to the two-state kinetic behavior of the protein. This first scenario is thus consistent with the initial barrier mechanism proposed by Krantz et al. [56]. The intermediate, even if it exists, is located in the native side after the free-energy barrier of folding (see Figure 6B). Such an intermediate is often called a hidden intermediate, and can be detected and characterized by the native-state H/D-exchange technique combined with 2D NMR spectroscopy [106]. In the second scenario, the backbone structure of the protein is organized by long-sequence-distant contacts, and hence there are too many ways to make the side-chain packing interactions quickly, keeping the second stage of the hierarchical folding rate-limiting, but destabilization of the intermediate leads to the two-state kinetic behavior. Although in this case, we apparently observe the two-state kinetics, a rigorous kinetic analysis may still follow Equation (2), and the apparently observed rate constant of folding (i.e., *k*_f_ in Equation (1)) should be equivalent to $(k_{\mathrm{UI}}/(k_{\mathrm{UI}}+k_{\mathrm{IU}}))\times k_{f}$ in Equation (2). The existence of an unstable high-energy intermediate was expected experimentally for certain two-state proteins from the unfolding-limb or the refolding-limb curvature of the chevron plot [107], and this is consistent with the two-state behavior in the second scenario. The free-energy profiles of the non-two-state and the two-state folding reactions are shown in Figure 6. Here, panels (A), (B), and (C) represent the hierarchical three-state folding, the two-state folding in the first scenario, and the two-state folding in the second scenario, respectively. From Figure 6, we can clearly see that the non-two-state folding reaction and the two-state folding reactions in the first and the second scenarios are closely related with each other, and a subtle change in the free-energy landscape of folding may result in a change in the apparent folding behavior. In fact, certain proteins that normally fold in the non-two-state manner can fold with two-state kinetics, depending on the solution conditions, such as changes in pH or temperature [108,109]. There are also cases in which mutations lead to the switching between two-state and non-two-state kinetics [110,111]. All these observations are reasonably understood in terms of the two-stage hierarchical model, so that the model may provide a unified picture of the mechanisms of protein folding.

## 5. Conclusions

1. In classic experimental studies on protein folding, the detection and characterization of kinetic folding intermediates were basic approaches to elucidate the molecular mechanisms of folding. The MG state was found and characterized as such a folding intermediate. The MG state in protein folding was thus briefly described. The characterization of the folding intermediate by time-resolved SAXS techniques, which was originally started by a collaboration with Ptitsyn’s group, led us to conclude that the intermediate thus characterized is equivalent to the MG state.

2. Due to the observation of the two-state folding behavior for a number of small globular proteins, the two-state folding engendered much interest among researchers in the field of protein folding. Apparently, the folding intermediate is not required for complete folding of two-state proteins, and this raised questions as to the significance of the folding intermediates, such as the previously observed and characterized MG state. This was especially the case for the burst-phase intermediate that accumulates within the dead time of conventional rapid reaction techniques such as the stopped-flow method.

3. The use of the continuous-flow techniques and more recently the in-gel folding techniques made it possible to directly observe the previously unobservable solution burst-phase process of folding. The solution burst-phase intermediate was shown to be a real folding intermediate separated from the U state by a free-energy barrier. Therefore, it is now clear that there are two types of globular proteins in terms of the folding behavior, i.e., two-state proteins and non-two-state proteins.

4. There are significant correlations between the logarithmic rate constant of folding, ln(*k*_f_), and various structure-based properties such as RCO, LRO, *Q*_d_ and so on. Although it was originally proposed that the correlation of ln(*k*_f_) with the structure-based properties exists only for two-state proteins, later it became clear that non-two-state proteins also show similar correlations with the structure-based properties, and hence the physical mechanisms behind two-state and non-two-state folding are essentially identical.

5. Finally, the two-stage hierarchical model was proposed as a general mechanism of protein folding. In this model, protein folding occurs in a hierarchical manner, reflecting the hierarchy of the native three-dimensional structure, as embodied in the case of non-two-state folding with an accumulation of the MG state as a folding intermediate. The two-state folding is thus merely a simplified version of the hierarchical folding caused either by an alteration in the rate-limiting step of folding or by destabilization of the intermediate. The two-stage hierarchical model also suggests that the non-two-state folding and the two-state folding are closely related with each other, and a subtle change in the free-energy landscape of folding may result in a change in the apparent folding behavior.

## Figures and Tables

**Figure 1 biomolecules-10-00407-f001:**
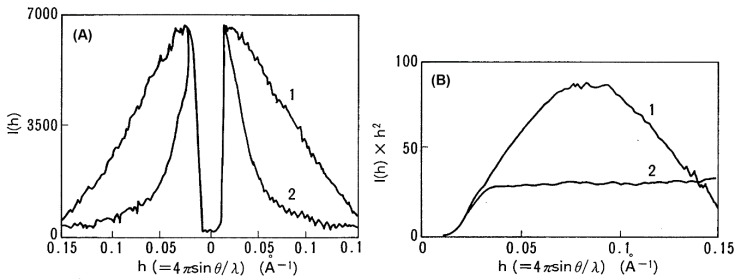
Synchrotron small-angle X-ray scattering (SAXS) patterns (**A**) and Kratky plots (**B**) for bovine carbonic anhydrase (1) in the N state (0.05 M Tris-HCl (pH 8)) and (2) in the U state (0.05 M Tris-HCl (pH 8), 8.5 M urea). Reproduced with permission from [28].

**Figure 2 biomolecules-10-00407-f002:**
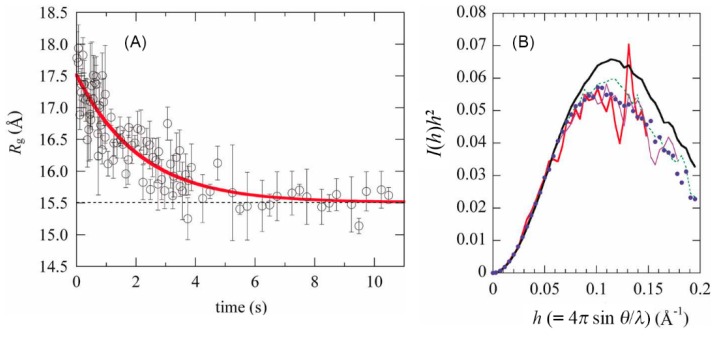
Time-dependent changes in the *R*_g_ during the refolding of α-lactalbumin (**A**), and the Kratky plots of the protein during the refolding (**B**). The refolding was induced by a concentration jump of GdmCl from 4 M to 0.77 M in the presence of 1 mM CaCl_2_ (pH 7.0 and 4.5 °C). (**A**) The time-dependent changes in *R*_g_^2^ were fitted to a single-exponential function, and the fitting curve of *R*_g_ thus obtained is shown as a thick red line. The dotted line shows the *R*_g_ value after the refolding is completed, and errors are the standard errors in fitting of Guinier plots. (**B**) The Kratky plots averaged between 10 and 20 ms after the initiation of refolding (red thick line), between 10 and 110 ms (purple line), between 10 and 960 ms (green dotted line), and between 8 and 11 s (black thick line). Blue filled circles show the Kratky plot of the equilibrium MG state at pH 2. The scattering intensities of each plot are normalized with the respective zero-angle intensities. Reproduced with permission from [31].

**Figure 3 biomolecules-10-00407-f003:**
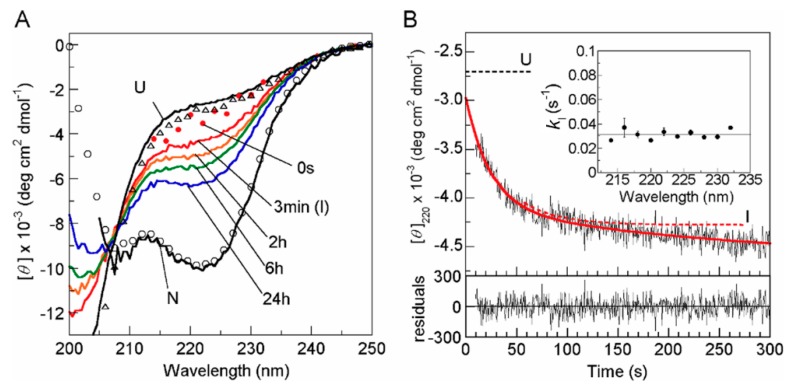
Representative far-UV CD spectra recorded during the folding of cytochrome *c* at 25 °C and pH 4.5 in a silica gel containing 80% (*w*/*w*) water. (**A**) The times indicated are those after initiation of the folding reaction. The red circles correspond to the mean residue ellipticity [*θ*] obtained by the extrapolation of kinetic curves to time zero. The solid black lines represent the spectra of the N and U states in gel. The open circles and triangles are the [*θ*] values of the spectra of the N and U states, respectively, in solution. Note that the spectra of the solution N and U states were recorded using the same folding and unfolding conditions as were used for the gel measurements. (**B**) A representative kinetic curve plotted using the [*θ*] at 220 nm for the initial resolvable cytochrome *c* in-gel folding phase at 25 °C. The times are those after initiation of the folding reaction. The thick red line is the fitted curve drawn using a multi-exponential decay function. The dotted red curve is the single exponential curve. The dotted black line in the upper panel is the [*θ*] at 220 nm for the U state. The residuals are shown in the lower panel. The inset shows the rate constants of the initial phase calculated for different wavelengths. Reproduced with permission from [16].

**Figure 4 biomolecules-10-00407-f004:**
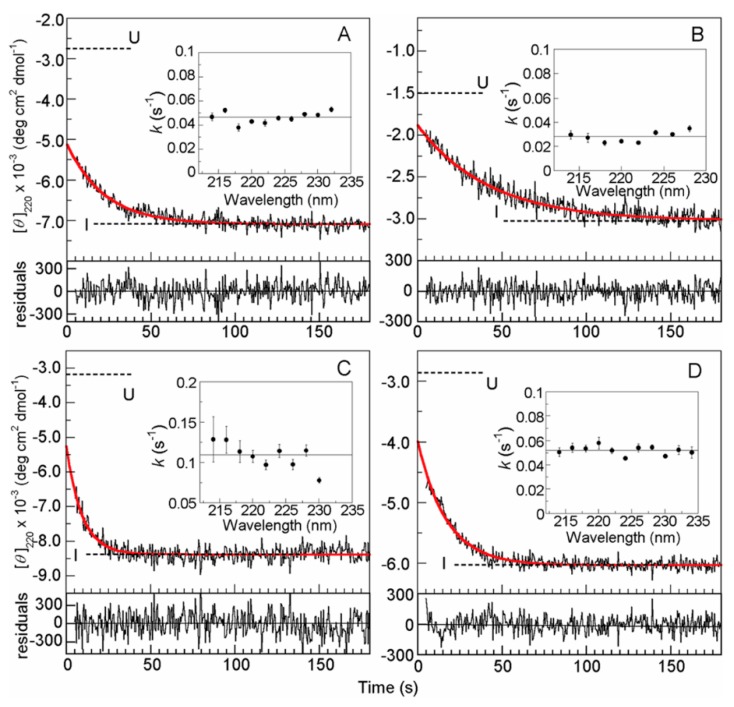
Representative plots of the [*θ*] at 220 nm during the U → I transitions at 25 °C for (**A**) equine β-lactoglobulin, (**B**) human tear lipocalin, (**C**) bovine α-lactalbumin, and (**D**) hen egg-white lysozyme in the gels. The times are those after initiation of the folding reaction. The thick red lines are each a fit Table 220. nm. The residuals are shown under each kinetic plot. The insets show the rate constants for the U → I phase calculated for each wavelength, with the horizontal lines being the average values. Reproduced with permission from [16]

**Figure 5 biomolecules-10-00407-f005:**
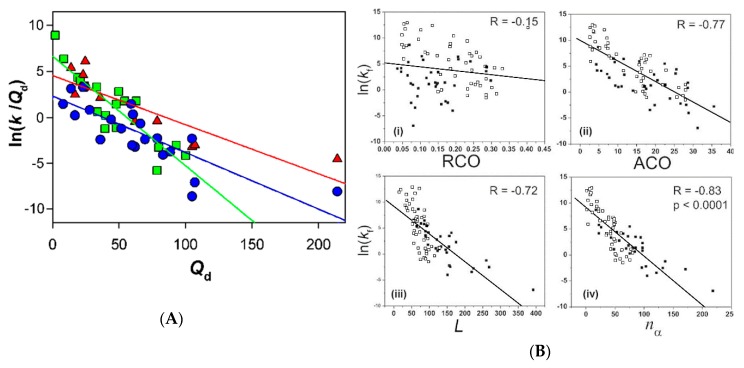
A comparison of the folding rates of two-state and non-two-state proteins. (**A**) The ln(*k*_f_/*Q*_d_) and ln(*k*_I_/*Q*_d_) values are plotted against *Q*_d_. Filled green squares (■) represent the ln(*k*_f_/*Q*_d_) values of two-state proteins. Filled red triangles (▲) represent the ln(*k*_I_/*Q*_d_) values of 10 non-two-state proteins for which the formation of the intermediate was kinetically observed. Filled circles (●) represent the ln(*k*_f_/*Q*_d_) values for 22 non-two-state proteins, i.e., the above 10 proteins plus the 12 proteins that show a burst-phase intermediate or only a roll-over in the chevron plot. The green, red, and blue continuous lines represent the best linear fit for ln(*k*_f_/*Q*_d_) for the 18 two-state proteins, ln(*k*_I_/*Q*_d_) for the 10 non-two-state proteins, and ln(*k*_f_/*Q*_d_) for the 22 non-two-state proteins, respectively. Reproduced and modified with permission from [89]. (**B**) Relationship between different structure-based properties and ln(*k*_f_) of two-state (open squares) and non-two-state (filled squares) proteins. (i) Relative contact order, RCO (*R* = −0.15); (ii) absolute contact order, ACO (*R* = −0.77); (iii) chain length, *L* (*R* = −0.72); and (iv) geometric contact number, *n*_α_ (*R* = −0.83), where *R* is the correlation coefficient. Reproduced with permission from [94].

**Figure 6 biomolecules-10-00407-f006:**
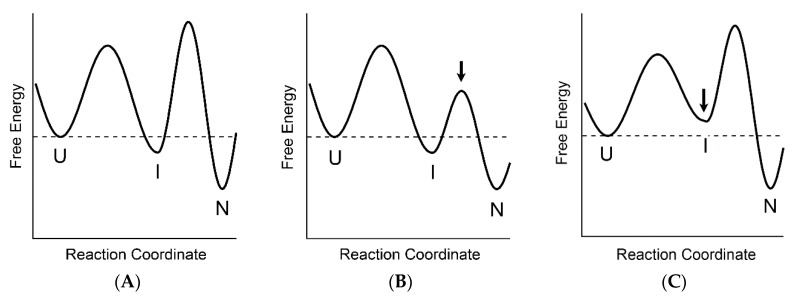
The free-energy profiles of the non-two-state and the two-state folding reactions. (**A**) Hierarchical three-state folding, (**B**) two-state folding in the first scenario, and (**C**) two-state folding in the second scenario. The horizontal broken line in each panel shows the free-energy level of U. Arrows in panel (**B**) and (**C**) indicate the places where the free-energy level has been changes as compared with the level in panel (**A**).

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
