# Peer review of "The Molten Globule, and Two-State vs. Non-Two-State Folding of Globular Proteins"

_biomolecules, 2020, doi:10.3390/biom10030407_

Round 1

Reviewer 1 Report

The author has provided an excellent review on the two-state and non-two-state folding behaviors. The manuscript is very well written and easy to follow. I only have minor comments listed below, most of which are interesting open questions that the authors can freely decide if they want to include in this review.

The argument that protein collapses before folding is still being revisited for the last couple of years, mostly due to the SAXS measurement that such initial collapse cannot be observed for some other proteins (Yoo et al, 2012 J Mol. Biol. 418:226). Whether water is a good or poor solvent for hydrophobic folded-protein-like sequence is still unclear (Riback et al 2017 Science 358:238). Since the authors reviewed their nice three state model to include both cases: 1) some proteins adopt an MG state with a smaller Rg than the unfolded state before folding and 2) for some others the MG state cannot be easily observed due to destabilization of the intermediate state, they might be able to provide their comments on this topic.

Temperature, pressure and force in addition to chemical denaturants are also commonly used for perturbing folding and and unfolding. The authors might provide comments how the picture of two-state/non-two-state folding can be learned in those cases.

The other interesting topic is if multiple experimental parameters are introduced, is the intermediate of folding always the same or can multiple folding pathways be observed in a higher dimension with more reaction coordinates (Guinn and Marqusee, 2019, Methods Protoc 2:32)?

There is increasing amount of interest on co-translational folding. The authors might provide comments on the applicability of these theories to the protein folding within the exit tunnel of the ribosome.

L121: 'Rg2' should read 'Rg'.

L170: It would be great to provide the PFDB link for easier access from readers.

Author Response

The argument that protein collapses before folding is still being revisited for the last couple of years, mostly due to the SAXS measurement that such initial collapse cannot be observed for some other proteins (Yoo et al, 2012 J Mol. Biol. 418:226). Whether water is a good or poor solvent for hydrophobic folded-protein-like sequence is still unclear (Riback et al 2017 Science 358:238). Since the authors reviewed their nice three state model to include both cases: 1) some proteins adopt an MG state with a smaller Rg than the unfolded state before folding and 2) for some others the MG state cannot be easily observed due to destabilization of the intermediate state, they might be able to provide their comments on this topic.

Thank you for the interesting comment. The protein studied by Yoo et al. (2012) is the B1 domain of protein L, a typical small two-state protein. Therefore, there is no MG state observed, and the compaction takes place concomitantly with the formation of N state. This is briefly decribed in the revised manuscript (lines 186-190)

Temperature, pressure and force in addition to chemical denaturants are also commonly used for perturbing folding and and unfolding. The authors might provide comments how the picture of two-state/non-two-state folding can be learned in those cases.

Following the comment, the reference to the pressure-induced MG state is provided (lines 97-98; ref. 19).

The other interesting topic is if multiple experimental parameters are introduced, is the intermediate of folding always the same or can multiple folding pathways be observed in a higher dimension with more reaction coordinates (Guinn and Marqusee, 2019, Methods Protoc 2:32)?

Thank you for an important comment. The multiple-pathway folding is now discussed in the revised manuscript (lines 329-332).

There is increasing amount of interest on co-translational folding. The authors might provide comments on the applicability of these theories to the protein folding within the exit tunnel of the ribosome.

This is an interesting point. However, the author feels that the co-translational folding is rather outside the scope of this article.

Reviewer 2 Report

The manuscript submitted by K. Kuwajiama is a vivid and brilliant summary of the state of the art in protein folding in vitro. It is very readable and a precious source of information not only for experts in the field but also for younger scientists and for newcomers in the field.

I only suggest some minor modifications to make the manuscript a bit more formal (and clearer).

Line 93. …, and Oleg proposed that…”: Maybe “Ptitsyn” might be better than “Oleg”.

Lines 106-107. “the Japanese group” might be called with its proper name.

Line 169. In the sentence “My colleagues in KIAS…” might be modified become “My colleagues in the Korea Institute for Advances Study…”

Line 172. “… and hence two-state …” might become “… and hence non-two-state …”

Author Response

Line 93. …, and Oleg proposed that…”: Maybe “Ptitsyn” might be better than “Oleg”.

Because this article is submitted to the special issue in memory of Professor Oleg B. Ptitsyn, the author would like to say "Oleg," because we called him in this way.

Lines 106-107. “the Japanese group” might be called with its proper name.

"the Japanese group" was change to "the Japanese members in our cooperative project."

Line 169. In the sentence “My colleagues in KIAS…” might be modified become “My colleagues in the Korea Institute for Advances Study…”

Thank you for a useful comment. This was done in the revised manuscript (line 190).

Line 172. “… and hence two-state …” might become “… and hence non-two-state …”

The two-state protein was thought to bee a rare example, when Jackson and Fersht reported the CI2 folding, but later many examples of the two-state protein were found. Therefore,  “… and hence two-state …” is correct.

Reviewer 3 Report

In the manuscript titled “The Molten Globule and Two-State vs. Non-Two-State Folding of Globular Proteins”, Dr. Kuwajima reviews our current understanding of the role of intermediates in protein folding, the likely molten globule nature of the common intermediates, and the experimental evidence for two-state and non-two-state folding behaviour in different proteins. The manuscript presents models for two-state and non-two-state folding through a molten globule intermediate, relates these models, and proposes a relationship to protein tertiary structure. The review is well-written and clear, and can serve as a useful primer for undergraduates and graduate students new to the field of protein folding kinetics, as well as a useful resource for later-career scientists seeking an overview of this topic. A mild criticism would be that the broad topics of protein folding mechanisms and kinetics have been covered by many past publications and reviews, though it can be difficult to find good reviews that focus on multi-state folding behaviour and the characterization of folding intermediates. I would suggest that some revisions would aid the comprehension of readers new to the field, but I believe that these should be doable without any major rewriting. My suggestions are enumerated below.

1. I am a bit concerned by the implication that each protein can be neatly divided into “two-state” or “non-two-state”. It has been shown that there are cases in which a protein that is observed by one method may appear two-state, but other methods may reveal subtler folding behaviour involving kinetic intermediates undetectable by the first method (see, for example, doi:10.1016/j.jmb.2008.08.024 and doi:10.1016/j.jmb.2011.04.027). It isn’t quite clear to me from Section 4 whether the assertion that “two-state folding is thus merely a simplified version of the hierarchical folding caused either by an alteration in the rate-limiting step of folding or by destabilization of the intermediate” is meant to imply that the intermediate is abolished entirely in a two-state folder (making it clearly distinct from a non-two-state folder) or whether this interpretation leaves open the possibility that more detailed experimental or analytical techniques could detect more complex folding behaviour associated with a poorly-populated intermediate despite the fact that certain experiments reveal only simple two-state behaviour. It would be helpful if this could be clarified.

2. The author assumes familiarity with certain concepts that would be understood by anyone with expertise in protein folding (and particularly in kinetic analysis). For readers new to the field, though, it may be worthwhile to provide a sentence at some point describing each, and a citation to allow readers to learn more. These concepts include:
a. A “state”. This is a concept that many primers on folding fail to adequately define for students of the field, leading many to picture, incorrectly, a well-defined single protein structure rather than a large ensemble of conformations of the protein and its surrounding solvent. Briefly clarifying early on that both the folded and unfolded states (and any intermediates, like the molten globule) are ensembles of microstates would help to provide a clearer picture for those new to the field. This would also set up the mention in Section 2 that the molten globule is defined by “heterogeneity of the three-dimensional structure” (a concept difficult to understand if the reader has not grasped that “state” does not mean “single rigid structure”).
b. Levinthal’s paradox. The first sentence of the second paragraph is accurate, but may confuse a reader new to protein folding kinetics. I would make it clearer what Levinthal’s paradox is, and that it was believed that a unique folding pathway could solve to the problem in that it provides a means by which the native state could be found, despite the vastness of the conformational space, in a shorter-than-astronomical period of time.
c. The concept that water is a poor solvent for unfolded polypeptides. This is mentioned but not explained in the third paragraph of Section 3 (“...since the water solution at a low concentration of GdmCl is a poor solvent for the polypeptides...”). This is a very important point for those new to the protein folding field, but one that can be counter-intuitive, and is therefore worth explaining.
d. Proline cis-trans isomerization. This is mentioned at the start of Section 3 before its relevance to folding kinetics is introduced in the third paragraph of Section 3, which could confuse a reader. (As an aside, why is it asserted that complex refolding kinetics caused by proline isomerization do not disqualify a protein from being considered a two-state folder in the first paragraph of Section 3? Wouldn’t a protein that stalls during refolding due to a slow cis-trans proline isomerization step be one with a kinetic intermediate, making it a non-two-state folder?)
e. Chevron plots. These are mentioned in the caption to Figure 5 and in the last sentence of Section 4, but are never introduced as the important tool that they are for analysing protein folding kinetics and for building protein folding models. A reader being introduced to two- and multi-state folding would probably benefit from a sentence or so of explanation before they are alluded to. If Figure 5 is kept (see the suggestion below that it be dropped outright), concepts like “burst-phase kinetics” and “rollover” would have to be explained, too (which might be difficult without a figure showing a chevron plot).
f. SAXS. One detail that would help a reader new to the field would be mentioning that SAXS gives information about the radius of gyration.
g. Circular dichroism. CD spectra are presented in Figure 3. A reader new to the field would benefit from a mention somewhere that a CD spectrum provides a fingerprint that reports on protein secondary structure.
h. Kratky plots. These are presented without explanation of what is being plotted with respect to what, which would only be meaningful to someone already familiar with this type of analysis (who likely wouldn’t need to read a review about protein folding).

3. I felt like the presentation sets up the expectation that this review would contain a bit of an historical perspective, particularly given the mention of the Human Frontier Science Program starting in 1993 in the introduction. As I read further, I felt a bit disappointed: only one other date is mentioned (at the start of Section 3, the discovery of two-state folding by Jackson and Ferscht in 1991). I would recommend either abandoning the historical perspective, or committing to it. My preference would be the latter: for those of us who came later to the field, we often get to read about the science as a set of established facts but rarely get to see it put in its context as a series of discoveries, so I would have welcomed more insight into when some of these discoveries were made, and by whom in collaboration with whom. (This can of course be inferred by looking at the references, but it’s appreciated to have it presented to the reader as a story.) I suspect that adding just a few more dates would give the manuscript a consistent feel of an historical narrative without requiring any major revision of the text.
a. As a minor point related to this, the final sentence of Section 2 mentions apparatus that was “used for nearly a decade”, but it isn’t clear from the context which era is being discussed without checking the references. Rough dates (e.g. “the early 2000s”) would help.
b. Also related to this, I feel like the conclusion summarizes the factual content of the paper, but doesn’t really present something that can be concluded from it. Could commentary on the development of these ideas about the kinetics of protein folding put more of a bow on it?

4. Some of the figures should probably be interpreted more for the reader:
c. Figure 1 shows SAXS patterns and Kratky plots, but the main takeaway for someone unfamiliar with these techniques is that the shape of these mysterious curves is different for the U and N states. Could these be interpreted in more detail for the reader? What is the “l(h)” that is being plotted, and what is the “h” that it is plotted against? Could the axes have lay-language labels instead of undefined notation?
d. I find Figure 5 to be more confusing than illuminating. Why is the folding rate normalized by Qd before being plotted against Qd? What are the sets of 17 and 22 two-state and non-two-state proteins that are referred to with the definite article (“the 22 non-two-state proteins”) without having been mentioned anywhere else? Why are there filled blue circles and open blue circles in panel A, and what is the significance of the difference, if any, of these sets? What is the significance of the slopes of the lines? It is asserted that this is due to a shift in the rate-limiting step, but how does one arrive at this conclusion? It feels like a jump in reasoning. Related to this, the expression for Qd is an unnecessarily mathematically precise means of saying what is stated much more simply in the next sentence: “Qd thus represents the number of sequence-distant native pairs” (separated by more than 12 residues). For a high-level review like this, I would stick to the intuitive English definition. All in all, it feels like Figure 5 is an unsuccessful attempt to summarize something very complex that was described in more detail in its original paper. I think that the figure should be simplified greatly or dropped outright.
e. Figure 6 could be clarified to indicate the feature that is changing in panels B and C. Perhaps this could be achieved with something as simple as an arrow pointing to the second kinetic barrier in panel B (drawing the eye to the fact that it has fallen), and to the intermediate well in C (drawing the eye to the fact that it has risen). A question, though: is the hierarchical model meant to imply that the intermediate disappears completely (i.e. that the second kinetic barrier drops below the energy of the intermediate in B, and that the intermediate rises above the energy of one of the flanking energy barriers in C)? If so, the figure should probably reflect this; if not, the text should probably make it clear that the intermediate still exists, albeit at sufficiently low levels or for sufficiently short periods of time that the folding appears to be two-state.

5. Although this manuscript is well-written overall, there were rare turns of phrase that I thought to be a bit confusing, or, in some cases, which may hinder comprehension by someone new to the field. These include:
a. “Traditionally the presence of a specific pathway of folding was thought to be a unique solution of Levinthal’s paradox.” The word “unique” confuses things a bit: it could be taken to mean that a folding pathway provides a specific and unique way to get from U to N in a reasonable amount of time, but could also mean that this is a particularly singular or creative idea about how proteins fold despite Levinthal’s paradox. I’d suggest rephrasing this sentence a bit. (See also my comment above about introducing Levinthal and commenting on how a folding pathway could solve the problem that it poses.)
b. “Oleg B. Ptitsyn, I and our colleagues” sounds odd to my ear -- I can’t recall seeing the first-person pronoun inserted between two third parties before. Perhaps “Oleg B. Ptitsyn, our colleagues, and I...” or “With our colleagues, Oleg B. Ptitsyn and I…”?
c. At the start of Section 3, I don’t think that it aids comprehension or is necessary for brevity’s sake to abbreviate “concentration” with “c”. Different people might think differently about this.
d. In the conclusion, the sentence “First, when the backbone topology is sufficiently simple and organized by short sequence-distant contacts, it is easy to determine the specific conformation of side-chain packing…” is a bit confusing. The phrase, “short sequence-distant contacts” is self-contradictory. Would “contacts between residues nearby in linear sequence” be clearer? Also, “easy to determine” could be mistaken to mean that it is easy for a human to predict the fold, not that the protein can find its own native state in a short time. Could this be made clearer?
e. Section 4 is mentioned in the introduction prior to Sections 2 or 3. This doesn’t seem necessary.

Author Response

Thank you for very careful reading of the manuscript and many valuable comments. Please see the attachment (Response_to_Reviewer3.pdf)
